# Multiple environmental stressors affect predation pressure in a tropical freshwater system
Costanza Zanghi [1] ✉, Iestyn L. Penry-Williams[1], Martin J. Genner [1], Amy E. Deacon[2] & Christos C. Ioannou [1]

Environmental change can alter predator-prey dynamics. However, studying predators in the context of co-occurring environmental stressors remains rare, especially under field conditions. Using in situ filming, we examined how multiple stressors, including temperature and turbidity, impact the distribution and behaviour of wild fish predators of Trinidadian guppies (*Poecilia reticulata*). The measured environmental variables accounted for 17.6% of variance in predator species composition. While predator species differed in their associations with environmental variables, the overall prevalence of predators was greatest in slow flowing, deeper, warmer and less turbid habitats. Moreover, these warmer and less turbid habitats were associated with earlier visits to the prey stimulus by predators, and more frequent predator visits and attacks. Our findings highlight the need to consider ecological complexity, such as co-occurring stressors, to better understand how environmental change affects predator-prey interactions.

Predators have a significant impact on their ecological communities. Through top-down effects predators can control the population size and distribution of prey species[1–3], and have non-lethal effects including changes in prey behaviour[4], cognition[5], growth rates[6], and reproduction[7]. Beside these effects, predation can influence species other than their prey by altering food webs and nutrient cycling[8–10]. Interspecific interactions are critical in freshwater habitats as these harbour high levels of biodiversity and population densities[11], yet species tend to be geographically constrained by physical barriers such as waterfalls, depth and salinity[12]. This implies that in freshwater habitats, prey species are often found in very close association to their predators[13].

Predation pressure, defined by predation frequency and prey vulnerability[14], is variable within a system and can depend on specific ecological and environmental conditions. Ecological factors impacting predation pressure include the abundances and distribution of predator and prey species[15,16], predators' search strategies, efficiency and prey selectivity[17–19], and intraspecific variation among predator individuals[20]. Additionally, environmental variables including temperature, flow rate, and water quality can have a significant impact on predation pressure[21–23]. Changes in water temperature can impact predation pressure by altering predators' activity levels, as higher energetic demands increase foraging behaviour[24]. Flow rate and dissolved oxygen can impact the distribution and behaviour of predators[25]. High flow rates can hinder predators' ability to

capture and handle prey[26], while low flow rates can result in poorer water quality[27,28]. Water quality, including factors such as dissolved oxygen and turbidity, can impact the physiology and survival of fish[29]. Increased turbidity can also alter sensory acuity, by acting as a visual barrier for predators who rely on vision to locate and capture prey[30,31]. Additionally, increased turbidity can affect the behaviour of the prey, for example eliciting greater antipredator responses[32,33] and altering their activity patterns (e.g., diurnal vs nocturnal[34]), with subsequent effects on predators' efficiency.

Predation pressure is thus dynamic, and sensitive to both biotic and abiotic factors. Natural systems are facing unprecedented levels of stress due to climate change and habitat degradation such as quarrying and urbanisation[35,36]. Human-driven rapid environmental change has significant negative impacts on freshwater systems[37]. These impacts can lead to declines in the abundance and distribution of predator fish species, as well as shifts in their behaviour[38–40]. Research investigating environmental effects on fish behaviour over the past decades has primarily focused on the impacts of individual environmental stressors; by environmental stressor here we refer to any environmental variable that can have a potential negative impact on natural systems or species[41] (e.g. heatwaves[42], water clarity[43], light pollution[44]). However, a variety of environmental stressors can simultaneously impact the behaviour and distribution of several species in natural systems[45,46]. Co-occurring stressors can lead to complex non-additive effects that are difficult to predict based on the effects of individual stressors

[1]University of Bristol, School of Biological Sciences, Life Sciences Building, 24 Tyndall Avenue, Bristol BS8 1TQ, UK. [2]The University of The West Indies, Department of Life Sciences, St Augustine, Trinidad and Tobago. ✉e-mail: cos.zan@gmail.com

considered in isolation[47–49]. For example, environmental stressors such as water temperature and salinity can interact antagonistically, beyond the sum of responses to the individual stressors, to mitigate interspecific aggressions and antagonistic behaviour in some fish species[50].

In this study we assess the effect of multiple environmental variables on the predatory behaviour of free-ranging, wild fish across seven rivers in the Northern Range, Trinidad and Tobago. This river system has been described as a "natural laboratory"[51] because of its multiple parallel independent rivers flowing down the mountain slopes, acting as natural replicates, while also being rich in biodiversity and accessible to researchers. These qualities have helped to establish a long history of experimental and observational work in this system. In particular, the Trinidadian guppy (*Poecilia reticulata*) has become one of the most studied model organisms in the fields of evolutionary ecology, genetics, and animal behaviour[52–57]. This small species is a widespread and abundant component of the fish assemblage in the Northern Range, and a valuable candidate to act as a prey stimulus for the diverse predator community in this system[58]. Alongside the collection of environmental data, we used in situ video recording to quantify predator species presence/absence and predator behaviours including the number of visits to the prey, time spent near prey and number of attacks on prey. Our main aim was to quantify associations between predation pressure and environmental conditions across sampling sites. Additionally, we aimed to assess the potential interactions between the environmental stressors recorded, as increased temperature and diminished water quality are prominent issues in freshwaters due to human activities[59,60]. We predicted that predation pressure (here measured as the time spent near the prey stimulus and the number of attacks) will be greater at higher temperatures due to the increased activity and foraging drive of fish[61]. However, we predicted that the effect of temperature on predation pressure will depend on water quality as, for example, predators may be delayed in approaching the prey due to a lack of visibility caused by turbidity[62,63] or prevented from engaging with the prey for longer periods of time due to high flow velocity or reduced dissolved oxygen[64,65].

Our results demonstrate a strong influence of environmental variables on the distribution of predators. Importantly, water temperature, dissolved oxygen and river flow velocity influenced the predatory behaviour of the observed species towards the presented guppies. Despite the absence of interactions among the environmental variables, they were all associated with the predators' presence to different extents. Taken together, this work represents the first broad-scale analysis of the influence of multiple

environmental variables on the predatory behaviour of wild, free-ranging freshwater fish through in situ observations.

## Results

### Environmental data

Some environmental variables were significantly correlated (Supplementary Fig. 1) and characterised the seven rivers differently (Supplementary Fig. 2). Sediment type was strongly associated with most other variables (Supplementary Fig. 3). Thus, to avoid issues associated with multicollinearity, sediment type was not included in the multivariate and behavioural analyses.

The eight remaining environmental variables were reliably captured by the first three principal components (each of which had an eigenvalue > 1[66]). We recognised variables with greatest contributions to a principal component using a loading threshold of > 0.35 or < −0.35. Light intensity and river width were positively associated with PC1, while canopy cover was negatively associated with PC1; temperature was positively associated with PC1, while flow rate and dissolved oxygen were negatively associated with PC2; river depth was positively associated with PC3, while turbidity was negatively associated with PC3 (Fig. 1a, Supplementary Table 2).

### Observed species

Fourteen species were identified in this study. Of these, two fish species were excluded due to rarity in the dataset (i.e. only observed once, with no attacks performed): *Corydoras aeneus* and *Synbranchus marmoratus*. An additional two species were excluded due to being herbivores: *Hypostomus robinii* and *Ancistrus maracasae*[67]. The only non-fish species recorded, the stream shrimp *Macrobrachium crenulatum*, was observed only in one site (Turure 1) and despite it being a known guppy predator[68], it was also excluded due to the experimental apparatus being designed for visual fish predators. The remaining eight fish species are potential guppy predators due to their carnivorous or omnivorous diet (Table 1), and combined, they were found near the prey stimulus for 58% of the trial time, as opposed to 21% of the time near the empty apparatus (the control treatment). Most predators were present across rivers: *Andinoacara pulcher* was ubiquitous, similarly *Saxatilia frenata* (previously known as *Crenicichla frenata*[69]) was only absent in the Guanapo river. The rarest species was *Roeboides dientonito*, only observed in the Aripo and the most downstream site in the Turure (Fig. 2a).

When leaf litter and pebbles were the predominant sediment type, the highest individual counts ( > 100) and number of species of predatory fish

(a)

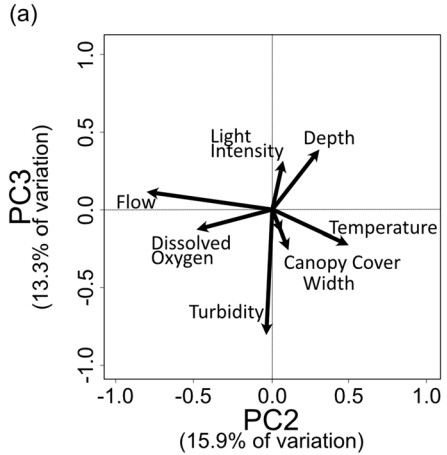

(b)

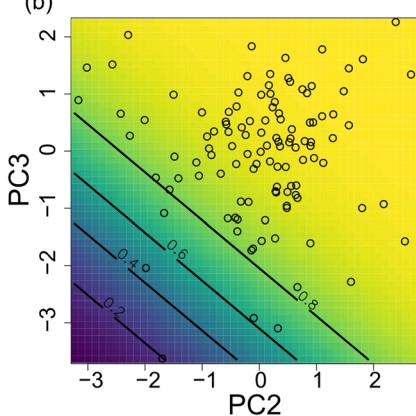

**Fig. 1 | Associations between environmental variables and their relationship with species presence/absence. a** Biplot for the Principal Component Analysis where the vectors represent the correlations between the PCA scores and the environmental variables. The plot is rotated to display PC2 and PC3 (instead of PC1 and PC2) to aid the interpretation of the second panel. **b** Model outcome for the predicted relationship between PC2 and PC3 for any predator species presence/absence, yellow represents greater species presence for values > 0. (positive values

for both PCs), while species absence is represented by dark blue at values < 0.2 (negative values for both PCs). Predictions are generated from the observed range for each PC from a GLMM with species presence/absence as the response variable and the main effects of PC2 and PC3 as explanatory variables. The black circles represent the raw data. The model also included a nested random effect of 1|River/Location.

## Table 1 | List of observed predator species included in the analysis

| | Scientific name | Family | Common Name | Diet[67] | Attack % | Attack rate |
|---|---|---|---|---|---|---|
| 1 | *Saxatilia frenata*\* | Cichlidae | Pike cichlid | Carnivore | 89.25 | 0.182 |
| 2 | *Astyanax bimaculatus* | Characidae | Two-spot Astyanax | Omnivore | 0.12 | 0.047 |
| 3 | *Hemibrycon taeniurus* | Characidae | Mountain stream sardine | Omnivore | 0.27 | 0.025 |
| 4 | *Andinoacara pulcher*† | Cichlidae | Blue acara | Omnivore | 7.65 | 0.025 |
| 5 | *Hoplias malabaricus* | Erythrinidae | Wolf fish | Piscivore | 0.31 | 0.023 |
| 6 | *Anablepsoides hartii*‡ | Rivulidae | Killifish | Carnivore | 2.13 | 0.016 |
| 7 | *Roeboides dientonito* | Characidae | Hunch back sardine | Carnivore | 0.27 | 0.005 |
| 8 | *Rhamdia* cf. *quelen* | Heptapteridae | River catfish | Omnivore | 0.00 | 0.000 |

Percentage of attacks performed by each species; the attack rate was calculated as the number of attacks divided by the time spent near the prey stimulus by each predator species.
Formerly known as: \**Crenicichla frenata* or *C. alta*, †*Aequidens pulcher*, ‡*Rivulus hartii*.

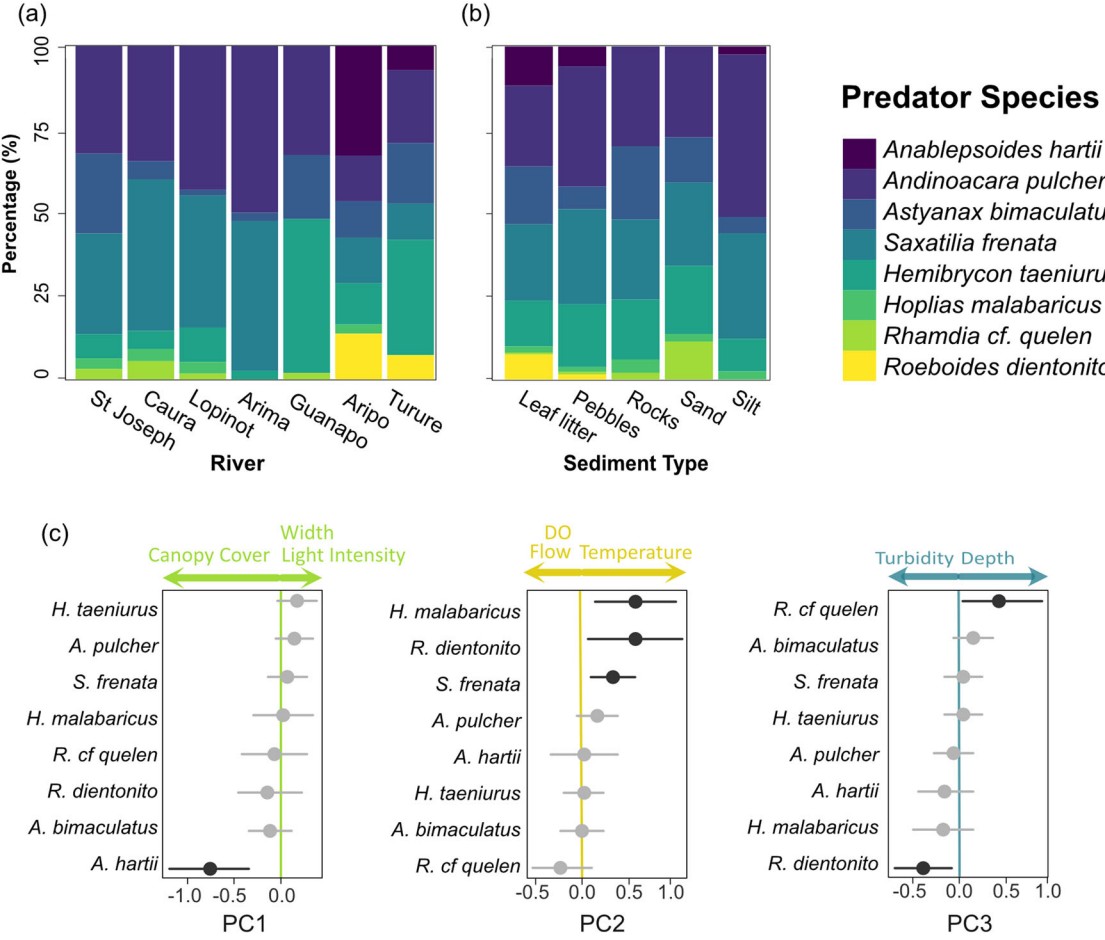

**Fig. 2 | Observed predator species.** Percentages of observed predator species across **a** rivers and **b** sediment types. **c** Estimated coefficients (circles) of the environmental variables and their 95% confidence intervals (lines) for the three principal components (PC) used as predictor variables in the GLLVM. The lines and circles coloured in grey denote intervals including 0, representing a lack of association between the species and the environmental variables. Species are ordered with the most positive coefficient for each PC at the top. Above each plot are reported the individual variables associated with the PC and the direction of the association.

were found, while rocky, sandy and silty substrates had the least number of species ( < 50 individual counts for each type, Fig. 2b). The only species not to perform attacks was *Rhamdia* cf. *quelen*, the seven other species were all observed to attack the guppies in the apparatus to different extents (Table 1). *S. frenata* was the most persistent predator with the greatest number of attacks (89% of all attacks, Table 1) and most time spent near the prey stimulus. After *S. frenata*, *A. pulcher* was the predator which proportionally performed the most attacks (7%), however it was also the most common predator found near the prey stimulus, ultimately having a lower attack rate than other species. *A. pulcher* was followed by *Anablepsoides hartii* and then *R. dientonito*, with the remaining species spending less than 15 minutes each in total near the guppies.

Guppy size (total length) varied across rivers (Supplementary Fig. 4), with larger individuals on average in the Caura and the smallest in the Guanapo (GLM, $R_2 = 0.22$, $F_{6,107} = 5.18$, $p < 0.001$). There was no significant effect of guppy size on predators behaviour (Supplementary Table 3).

**Table 2 | Model comparisons of the five behavioural response variables for the prey stimulus treatment**

| 1 Presence/Absence | ΔAICc | df | 2 Time first visit | ΔAICc | df | 3 No. Visits | ΔAICc | df |
|---|---|---|---|---|---|---|---|---|
| PC2*PC3 | 0.0 | 5 | PC2 + PC3 | 0 | 6 | PC2 + PC3 | 0 | 6 |
| PC2 + PC3 | 1.6 | 4 | PC2 | 0.9 | 5 | PC2 | 0.4 | 5 |
| PC3 | 3.7 | 3 | PC2*PC3 | 1.7 | 7 | PC2*PC3 | 1.1 | 7 |
| PC2 | 5 | 3 | PC1*PC2 | 3 | 7 | PC1 + PC2 | 2.3 | 6 |
| PC1 + PC3 | 5.8 | 4 | PC1 + PC2 | 3.1 | 6 | PC1*PC2 | 4.1 | 7 |
| Control first | 7.2 | 3 | Time of day | 9.1 | 5 | **Null** | **22.5** | **4** |
| **Null** | **7.4** | **2** | PC3 | 9.8 | 5 | Replicate | 23.5 | 5 |
| Time of day | 7.4 | 3 | Replicate | 10 | 5 | PC3 | 23.8 | 5 |
| PC1 + PC2 | 7.5 | 4 | **Null** | **10.5** | **4** | Control first | 24.3 | 5 |
| PC1*PC3 | 8.6 | 5 | Control first | 11.6 | 5 | PC1*PC3 | 24.4 | 7 |
| Replicate | 8.6 | 3 | PC1 + PC3 | 11.6 | 6 | PC1 | 24.6 | 5 |
| PC1 | 9.4 | 3 | PC1 | 12.2 | 5 | Time of day | 24.7 | 5 |
| PC1*PC2 | 9.5 | 5 | PC1*PC3 | 13.8 | 7 | PC1 + PC3 | 25.9 | 6 |

| 4 Time near stimulus | | | 5 No. Attacks | | | |
|---|---|---|---|---|---|---|
| Replicate | 0 | 6 | PC2 | 0 | 5 | |
| PC2 | 0.4 | 6 | PC2 + PC3 | 1.7 | 6 | **PC1:** - Canopy cover + Light intensity + Width |
| **Null** | **0.6** | **5** | PC1 + PC2 | 2.2 | 6 | |
| PC1 + PC2 | 1.9 | 7 | PC1*PC2 | 3.6 | 7 | |
| PC3 + PC2 | 2.1 | 7 | PC2*PC3 | 3.8 | 7 | **PC2:** - Flow + Temperature - Dissolved Oxygen |
| PC3 | 2.4 | 6 | Time of day | 4.7 | 5 | |
| PC1 | 2.7 | 6 | **Null** | **5.2** | **4** | |
| Control first | 2.7 | 6 | PC1 | 6 | 5 | **PC3:** - Turbidity + Depth |
| Time of day | 2.8 | 6 | PC3 | 6.5 | 5 | |
| PC2*PC3 | 4.1 | 8 | Replicate | 6.7 | 5 | |
| PC1*PC2 | 4.1 | 8 | Control first | 7 | 5 | |
| PC1 + PC3 | 4.6 | 7 | PC1 + PC3 | 7.8 | 6 | |
| PC1*PC3 | 6.2 | 8 | PC1*PC3 | 8.9 | 7 | |

Each row represents an individual model varying in the explanatory variables: the plus sign indicates multiple variables are main effects, while the asterisks indicate an interaction term between the principal components (in addition to their main effects). The first set of models (presence/absence) include the sampled river as random effect. The other models include a nested random effect of 1| River/Location. ΔAICc: Difference in the small sample corrected AIC between each model and the most likely model. Df Number of components for each model. In bold are highlighted the null models' scores used in the comparison. In the bottom right corner are detailed the individual variables contributing to each PC.

## Environmental predictors of predator species composition

The GLLVM demonstrated significant effects of the environmental variables on the presence/absence of most of the predator species (Fig. 2c). Higher values of PC1 (indicating less canopy cover, wider river width, and greater light intensity) were associated with a reduced occurrence of *A. hartii* (p < 0.001). Higher values of PC2 (indicating reduced flow, high temperature and low dissolved oxygen) were associated with greater occurrence of *H. malabaricus* (p = 0.009), *R. dientonito* (p = 0.028) and *S. frenata* (p = 0.005). Greater values of PC3 (indicating reduced turbidity and increased depth) were associated with greater occurrence of *R. quelen* (p = 0.049), but a lower occurrence of *R. dientonito* (p = 0.02) (Supplementary Table 4).

The CCA revealed that CCA axis 1 captured 7.3% of the explained variance, while CCA axis 2 accounted for 4.6% (Supplementary Fig. 5). Together, the environmental variables considered in the analysis accounted for 17.6% of the variance observed in the presence/absence data. Specifically, there were significant effects for flow rate (ANOVA; $F_{1,97} = 4.423$,

$p = 0.003$), river width ($F_{1,97} = 3.424$, $p = 0.008$), turbidity ($F_{1,97} = 3.144$, $p = 0.012$) and temperature ($F_{1,97} = 4.315$, $p = 0.004$).

## Predator presence

The most important predictors for the presence of predators were PC2 (flow rate, temperature and DO) and PC3 (turbidity and depth), where there was a marked decrease of predator abundance in turbid and shallow sites (PC3) when sites were also cooler with high flow and dissolved oxygen (PC2; Fig. 1b). The pattern in Fig. 1b shows that predators are more likely to be present as either PC increases gradually. Additionally, it shows a sharp transition between species presence and absence, suggesting that once certain environmental conditions are met for one PC or the other, the habitat becomes suitable for these predators. The influence of co-occurring PC2 and PC3 was observed when the prey stimulus was present, but when guppies were absent, PC2 was found to be less important (Table 2(1), Supplementary Table 5.a). The presence of guppies did not predict predators' presence/absence, as the model including treatment type was less likely than the null model (Supplementary Table 5a, Supplementary Fig. 6).

## Predator behaviour

PC2 was the leading environmental predictor for most other measures of predator behaviour, where increased temperature, lower dissolved oxygen and slower flow were associated with earlier and an increased number of predator visits to the prey stimulus (Fig. 3a, b), and more attacks on prey (Fig. 3d). The time of the first predator visit to the apparatus was driven by both PC2 and PC3 however, the model including PC2 was the most likely for the dataset which only included the prey treatment (Table 2(2), while the model that included only PC3 was the most likely for the control treatment dataset (Supplementary Table 5.b). The total number of visits to the prey stimulus was strongly driven by PC2 (Fig. 3b, Table 2(3). For the dataset including the prey treatment only for the number of attacks performed, the model including PC2 was the most likely (Table 2(5). The presence of prey guppies was also an important predictor, showing that when guppies were present, more visits occurred (Supplementary Table 5.c, Fig. 3b). The presence of the prey stimulus was the key driver of predator engagement with the apparatus, as it was the strongest predictor of the total time spent near the stimulus and number of attacks performed (Table 2(4), Supplementary Table 5.d, e, Fig. 3c, d).

## Discussion

Through the observation of free-ranging predators' behaviour in the wild, this study provides evidence that the co-occurrence of multiple environmental variables has the potential to shape fish communities and predator behaviour, and hence predator-prey interactions. Flow rate, temperature and dissolved oxygen (PC2) all affect important physiological processes in fish and can impact their activity and motility[39,70]. These parameters were the most important predictors of predator behaviour in our study, with evidence that they affect multiple stages of the interaction between predator and prey, from encounter to attack[71]. Specifically, warmer temperature, lower dissolved oxygen and slower flow were associated with predators approaching the prey earlier and more often, and attacking them more often. Together, this suggests that guppies would face greater predation risk under these environmental conditions.

The lack of a treatment effect on predator presence and the time of first visit suggests that the predators were not visually locating prey from a distance and then approaching the apparatus. Rather, depending on water quality, predators were observed near the apparatus regardless of treatment. However, they would remain near the stimulus for longer and make more visits and attacks when guppies were present. Despite canopy cover and light intensity varying widely across sites in our system, brightness was not found to impact the predators' behaviours. Predator presence increased with lower turbidity, greater depth, slower flow, warmer temperature and reduced dissolved oxygen. This suggests that, rather than impacting fish sensory acuity (i.e. vision), these parameters affected their distribution[26,72,73], strongly

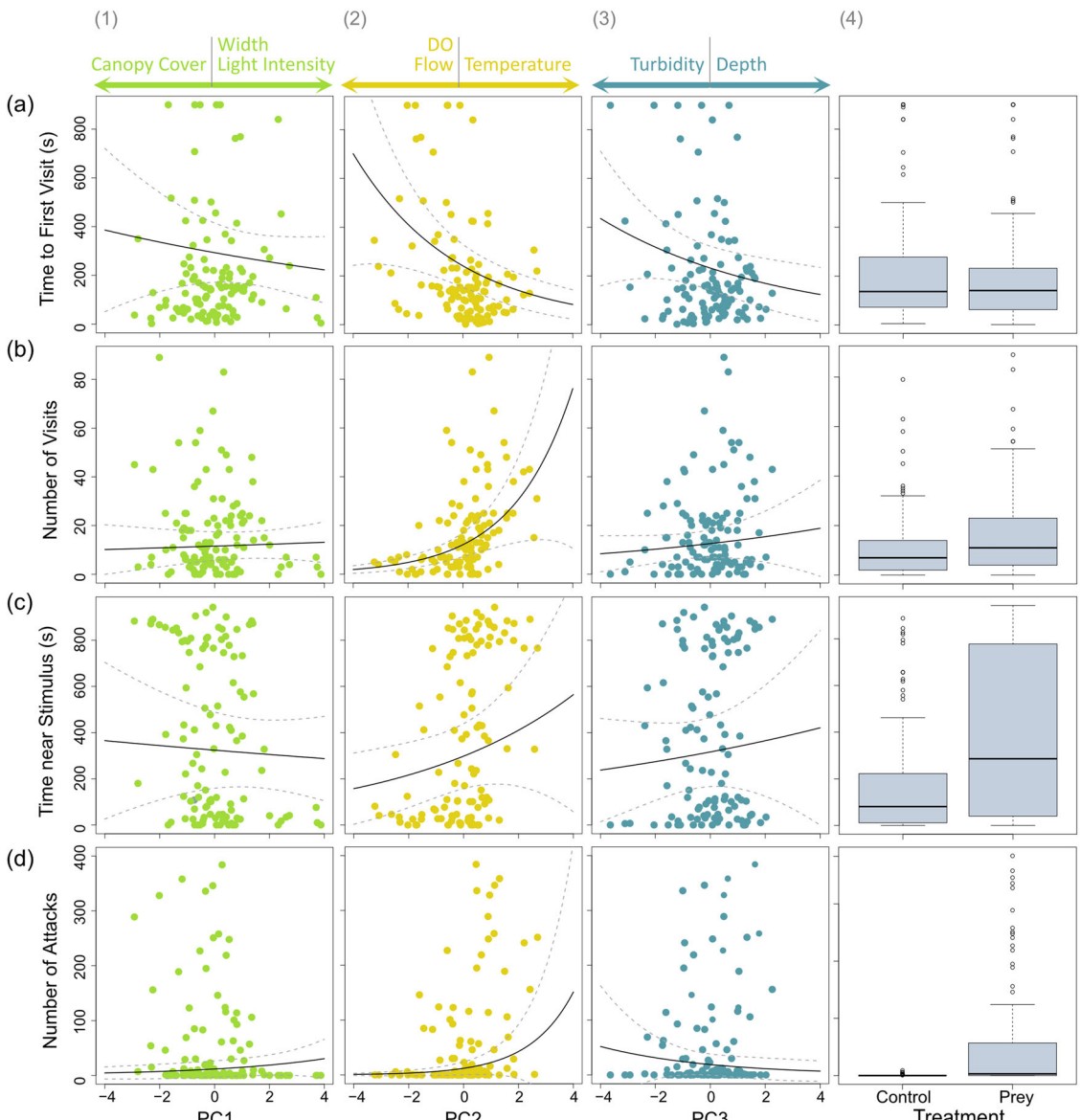

**Fig. 3 | Generalised linear mixed models (GLMMs) of associations between environmental variables and the behaviours of predators.** In each scatterplot there are 114 sampling events from the prey stimulus treatment. **a** Time taken until the first visit to the prey stimulus, **b** number of visits to the prey, **c** time spent near the prey, **d** number of attacks on the prey, all in relation to PC1 (1), PC2 (2) and PC3 (3). The black lines represent the predicted values calculated from GLMM coefficient, while the dotted grey lines represent the 95% confidence intervals. Above the plots are reported the individual variables associated with the PC. In the fourth column, treatment (i.e. control vs prey stimulus) effect on the behavioural responses ($N = 228$). Horizontal black lines within the boxes represent the median value. The edges of the boxes represent the lower (25th percentile) and upper (75th percentile) quartiles. The whiskers extend from the most extreme data point by 1.5 × the interquartile range. The black circles represent outliers.

influencing the time at which predators were first observed. Predators were near the apparatus for more than half the recorded time when prey were presented. However when present, different species represented different levels of risk (as shown by the attack rate)[58]; for example, *S. frenata* would perform numerous repeated attacks[20], *H. malabaricus* would often approach the recording apparatus slowly and leave after performing a single attack, while *A. pulcher* would spend a long period of time near the prey without performing an attack (Supplementary Video).

In addition to these behavioural differences between species, their presence/absence was differentially affected by the environmental parameters. *A. hartii* favoured narrower and darker stretches of river, *H. malabaricus*, *R. dientonito* and *S. frenata* were predominantly associated with sites with higher temperature and less flow and dissolved oxygen, while turbidity and river depth had contrasting effects on the presence of *R. cf quelen* and *R. dientonito*, the first more associated with clearer and deeper

water and the second with shallower and more turbid sites. While it is important to note that this study does not demonstrate causality, our observations are supported by previous work carried out in this system[55,57,58], and highlight potential relationships between the observed variables and investigated behaviours.

Considering the species-specific differences in response to environmental variability is important as they imply that predicting species abundances or local extinctions due to rapid environmental change becomes challenging when assuming uniform responses from all predator species. This is true especially if predictions are based on overly simplistic systems, where only individual parameters, a reduced number of species with single sensory modalities of foraging[74] are taken into consideration. Ultimately, the predation risk experienced by the guppies may shift from visual fish predators to other taxa, for example invertebrates, which have been found to be more resilient to shifts in temperature and turbidity[75]. To further this work,

trials exploring non-fish predators such as the stream shrimp *M. crenulatum*, fishing spiders, and aerial predators such as bats and birds, as well as crepuscular and nocturnal trials would be important to gain a more thorough understanding of intraspecific interactions at a wider scale. Future work should include differences in prey abundance, as this can have consequences for predator's presence and behaviour. Similarly, as increased temperature and turbidity interact to reduce guppy shoal cohesion in the presence of blue acaras[76], future work could focus on prey's predator avoidance and escape responses under complex environmental conditions and a multi-predator species scenario[77–80]. Similarly, our experimental approach could be used to investigate other types of intra- and interspecific interactions. For example, it could explore aggression and competition among predators[20], predators' abiotic niches, as well as the effects of guppy presence on the social foraging behaviour of free-roaming guppies and other non-predator species[81–83].

While in this study PC2 and PC3 both had an influence on the presence/absence of observed predators, PC2 alone was the leading predictor for most behaviours assessed. The lack of interactive effects observed may be explained by the sensitivity of fish to the different environmental conditions at the time of testing. Specifically, the range of variable values observed at the field sites (e.g. mean temperature variation of 5.6 °C, 13 NTU for turbidity, or 9.8 mg/L for dissolved oxygen) may not have been wide enough to elicit significant interactions between the variables. Additionally, as the observations were conducted at the end of the dry season, a level of seasonal adaptation among the fish populations may have also occurred, leading to less variable behavioural responses. Our results provide a snapshot on the influence of current environmental variability on predator behaviour in the wild; to be able to assess and predict responses to future or impacted conditions (i.e. out-of-season weather events) further investigations should incorporate a broader environmental variability with repeated sampling through the year and complementary laboratory experiments. This approach could help to further untangle relationships between co-occurring environmental variables and their influence on focal behaviours.

The findings of this study highlight the association between contemporary environmental change and the key variables influencing predator species distribution and behaviours. Altered rainfall patterns due to climate change impact water temperature and river flow[84,85], while human activities such as farming, deforestation, and quarrying contribute to soil erosion, leading to sediment runoff into waterways and subsequent eutrophication, turbidity, and hypoxia[37,86,87]. Within the complexity of such systems, recognising general trends is important, but it is equally crucial to identify how specific variables and their potential interactions can influence natural populations. Assessing the impact of multiple environmental stressors on animals and plants is challenging due to the large number of potential stressors and the complexity of their interactions[11,49], but it can be achieved through a combination of experimental design, monitoring, modelling, and data synthesis. In contrast to their prey, studies on the behaviour of predators in the wild are particularly rare because of the lower density of predators and the unpredictability of predation events. The Northern Range in Trinidad provides an invaluable study system in which predators can be presented with prey stimuli and their responses measured, with substantial variation in environmental variables and predator species communities over a small geographical area. In future work, researchers can approach this topic by designing laboratory experiments that manipulate stressors under controlled conditions[42,88–90]. This allows for the isolation of the effects of specific stressors and evaluation of their interactions with other stressors. Importantly, these laboratory studies should be linked to field observations as researchers can monitor changes in animal behaviour and distribution in response to current changes in environmental conditions. The aggregation of stressors that are often closely associated in natural and modified systems not only allows for simplified statistical analyses but can potentially highlight trends and dominant drivers[91]. This provides valuable information on the in situ impacts of multiple stressors on communities

with practical implications for effective management of natural resources at a local scale[92]. This ground truthing can then be applied in agent-based models to simulate the impacts of different combinations of stressors and predict their effects under different scenarios[93]. Finally, data can be synthesised from multiple sources to identify patterns and trends in the impacts of multiple stressors on populations and ecosystems[94]. This information is critical for understanding the impacts of environmental change on communities and ecosystems, and for developing effective conservation strategies to protect biodiversity and preserve ecosystem services[95–97].

## Methods

### Study Sites

Environmental data and behavioural observations were obtained from 19 sites across 7 rivers in the Northern Range of Trinidad (Fig. 4a). The sites were selected to be a minimum 400 m distance from one another, having safe accessibility, and the presence of guppies, which was assessed by performing 10 min of continuous seining. To achieve a representative sample size, three sites per river were identified, except for the Guanapo and Turure rivers, where only two sites met the selection criteria. Within each of the 19 sites, sampling was carried out at three locations, for a total of 57 sampling locations (Fig. 4b). These locations had a minimum depth of 40 cm to allow recording from a top view camera, and were a minimum of 5 m from one another to reduce the chance of recording the same individual predators (Supplementary Table 1). Each sampling location was sampled twice with a minimum of 4 days between repeated visits, for a total of 114 site visits. Repeated sampling was alternated for time of day (am or pm) and treatment type (control or prey stimulus) to account for diurnal variation in fish activity levels and response to a novel object (the recording apparatus). Each day, two sites were visited, one in the morning and the other in the afternoon. The first site of the day and treatment was randomised; the second site was selected as the closest to the first to minimise travel time and ensure sampling could be completed in one day. After all 19 sites were visited once, the second sampling block was repeated, this time alternating time of day and treatment order from the first sampling block. There were two exceptions to this balanced design due to weather conditions: downstream Aripo and midstream Arima were sampled in the morning in both sampling blocks. Fieldwork was carried out at the end of Trinidad's dry season (April-May 2022).

### Environmental parameters

During each site visit ($N_{Visit}$=114), nine environmental parameters were recorded where the recording apparatus was positioned to capture a snapshot of the environmental conditions at the time of the behavioural observations. River width and depth were recorded (cm) with a tape measure once per site visit (width) or once per sampling event (depth) at the location where behavioural observations were carried out. Such locations were haphazardly selected points which had a minimum depth and width of 40 cm to allow for underwater video recording. Temperature (°C) and light intensity (lux) were measured with an in-water sensor (HOBO-MX2202) positioned on the riverbed recording one measurement for each parameter every 30 s for a continuous 15 min period during each sampling event. Turbidity (nephelometric turbidity unit, NTU) was measured from three 5 ml discrete water samples with a turbidity meter (Thermo Scientific Orion AQUAfast-AQ3010) at each location per sampling event. Dissolved oxygen (mg/L) was measured with an in-water sensor (Lutron DO Meter PDO-519) placed 10 cm under the water surface once per sampling event. Flow rate (m/s) was measured once per sampling event with a flowmeter (GEO-PACKS-MFP51) placed 15 cm from the riverbed. A wide-angle photo of the canopy above the recording apparatus was taken at each site visit. Using image analysis software (imageJ[98]), the blue channel of each photo was converted to a binary mask using the Yen threshold method[99] to obtain a measure of vegetation canopy cover (%). The predominant sediment type (leaf litter, rock, pebbles, sand or silt) for each location was manually scored from a photo taken at 40 cm from the riverbed.

**Fig. 4 | Sampling regime in the Trinidadian Northern Range, Trinidad & Tobago. a** Map of the 19 field sites across seven rivers. **b** Example of the three locations within each site. **c** Control (above) and prey stimulus (below) treatments for each location. **d** Examples of matched video frames from top and side-view videos.

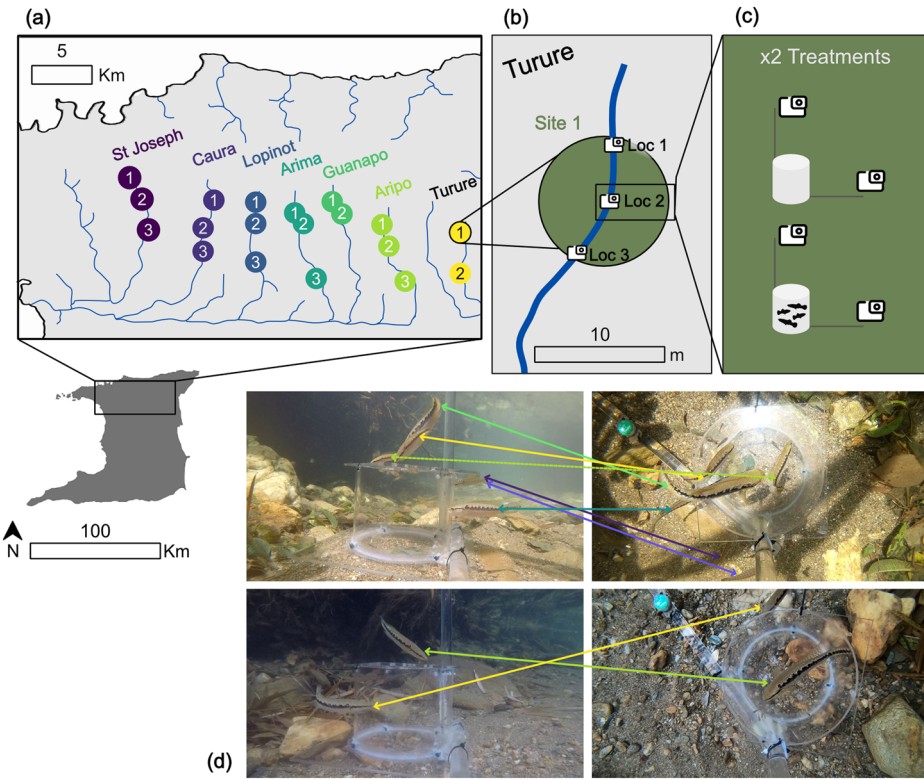

## Behavioural observations

At each sampling location, an apparatus carrying a side and a top view camera (GOPRO HERO6) was deployed to record the behaviour of free-ranging diurnal predatory fish. The cameras were angled at 90° from one another and placed 40 cm from the centre of the apparatus, and filmed at 1080p resolution, 30 frames per second and wide field of view. The apparatus consisted of a cylinder (Ø = 20 cm, H = 15 cm), with a base and removable lid, all made from transparent acrylic glass[20]. In addition to sampling each location twice on different days, at each visit to a site, the apparatus was deployed twice at each of the 3 locations (Fig. 4c), with a minimum of 30 min between repeated deployment in the same location, for a total of 228 sampling events. The apparatus was either deployed empty to act as a control, or with guppies enclosed within the cylinder to act as a prey stimulus to the predators.

On each site visit before sampling started, guppies were collected with seine nets a minimum of 5 m downstream from the most downstream location at each site to minimise disturbance to the fish and sediment. Fish collection and handling was approved by the Ministry of Agriculture, Land and Fisheries (Trinidad and Tobago) and performed in accordance with ASAB/ABS guidelines for the treatment of animals in behavioural research. The research was approved by the University of Bristol Welfare and Ethical Review Body (UIN/21/003). Once collected, guppies were housed for 20 to 80 min in a temperature-insulated container (20 × 30 cm) with water collected from the sampled site, before being enclosed in the recording apparatus for the prey stimulus treatment. These fish were used as a stimulus to the free-ranging predators at all three locations within that site visit, and no behavioural observations were made on the guppies. The prey stimulus consisted of a mixed-sex shoal of 10 adult guppies (2 males and 8 females where possible, mean total length = 24.5 mm ± 2.9 SD). The inclusion of both male and female guppies in a relatively large shoal was intended to present a more distinct and easily recognisable stimulus to predators. To deploy the apparatus, each location was approached from downstream to minimise disturbance to the fish and resuspension of sediment. Once in place, the cameras recorded for 15 min.

## Behavioural data extraction

The side and top view video recordings from the 228 sampling events were manually synchronised and scored using BORIS v.7.9.8[100] (Fig. 4d). To be included, each fish was required to enter the view of the top camera at least once and it needed to be identified to species level. Individuals were not identified between visits (as done by Szopa-Comley and colleagues[20]), therefore each fish entering the top-camera view was recorded as a new individual, unless it was clear from the side-camera view that it was the same individual (e.g. a fish entering the top-camera view multiple times was counted as the same individual if it remained within view of the side camera). This information was used to compile presence/absence data for each species at each site. The ethograms produced with BORIS were subsequently analysed in R v.3.6.1[101] to extract the following behavioural metrics at species level: the total number of visits, i.e. every occasion in which a fish entered both top and side-camera views; the amount of time spent in proximity of the cylinder, calculated as the 'exit' minus the 'enter' times of each fish from the top-camera view; the time of the first visit, i.e. the time at which a fish entered the top-camera view for the first time; and the total number of attacks, calculated as the sum of predatory events, classified as a fish performing a tap to the cylinder with its mouth.

## Statistics and reproducibility

**Environmental data.** Each variable was averaged (mode for sediment type, mean for all others) for each sampling event ($N_{Obs}$=228). To stabilise the variance of the averaged data, continuous environmental variables were $\log_{10}(x + 1)$ transformed. All continuous variables were checked for collinearity (Spearman multivariate correlations with the pairs.panels() function in the R package 'psych' v.2.3.3[102]). All environmental variables were also compared across rivers with Kruskal-Wallis tests, followed by pairwise Dunn's tests with the Benjamini-Hochberg correction for multiple comparison (R package 'stats4' v.4.3.0[101]). The presence of an active rock quarry any distance upstream from each site was recorded. This was used to assess if quarry presence was associated with increased turbidity (Mann-Whitney U test) and sediment siltation

(Chi-Square test) in the sites further downstream, both using the R package 'stats4' v.4.3.0[101].

To reduce the dimensionality of the environmental data, a Principal Component Analysis (PCA) was performed using the prcomp() function which centres and scales the data prior to analysis (R package 'stats4'[101]). The Principal Component axes with eigenvalues > 1 were retained as predictor variables for the multivariate and behavioural analyses[103].

**Observed species.** To quantify the relative risk posed by different predator species, a predation score was calculated as the rate of attacks per unit time. This was obtained for each species by dividing the total number of attacks on guppies by the total time spent near the prey stimulus. Subsequently, each predator species was ranked in comparison to the others.

All guppies used as the prey stimulus were measured (total body length). To assess whether guppy size influenced the behaviour of the predators, all five behavioural metrics (predators' presence, the total number of visits, the time of first visit, the time spent near the stimulus (both times rounded to the nearest second) and the total number of attacks) were tested as a function of guppy total length with generalised linear mixed models (GLMM; 'glmmTMB' v.1.1.7 package in R[104]). The GLMMs, were based on the subset of the data which only included the prey treatment (N = 114). We included the random effects of location nested in river; we excluded the site as a random effect to reduce the complexity of the models. Model residuals were checked for over-dispersion using the 'DHARMa' v.0.4.6 package in R[105]. The model for presence/absence was fitted using a binomial distribution, while all other models were fitted using a negative binomial distribution. To resolve model's convergence issues, zero inflation was included for the time spent near the stimulus. Likelihood ratio tests were then used to assess the statistical significance of guppy size on the five predators' behaviours (drop1 function in 'stats' v. 4.3.0 package in R[106]), and the effect size was estimated according to Brysbaert and Stevens[107].

**Multivariate data.** The impact of the environmental variables on each predator species' presence/absence was analysed with a generalised linear latent variable model (GLLVM; 'gllvm' v1.4.1 R package[108]). GLLVMs can be powerful tools that allow integration of multiple predictor variables into a single model, while accounting for differences in the response of multiple species[109]. Due to the binary response (presence/absence), the model was fitted as a binomial distribution, while the predictor variables were the principal components retained from the PCA performed on the environmental data. We systematically tested the model with varying numbers of latent variables. For each model, we assessed the goodness of fit using the Akaike Information Criterion (AIC)[108]. According to these scores, the best fitting model included one latent variable. The fit of the model was checked using residual diagnostic plots: randomised quantile-based residuals plotted against linear predictors, and QQ plots (plot(-model) function in 'gllvm' package). The statistical significance of the PC association with each fish species was calculated using the model summary() function.

Additionally, a Canonical Correspondence Analysis (CCA) was performed to quantify the association of all environmental variables ($\log_{10}(x + 1)$ transformed) with each predator species across all sampled rivers[110]. The variance explained by each environmental variable was then quantified using the anova.cca() function in the 'vegan' v2.6.4 R package, with the "by margin" option and 100,000 permutations[111].

**Behavioural data.** The influence of the environmental variables on each behavioural metric was assessed using GLMMs[104]. For each behavioural response variable (i.e. any predator presence/absence, total number of visits, time to first visit, time spent near stimulus and total number of attacks), a null model was constructed including only the response variable and the random effects of sampling location nested in the river. Each null model was compared to 13 models with the same response variable, each of which included an additional predictor or a two way-interaction between predictors (and their main effects). The explanatory variables included in the analysis as predictors were: the principal components retained from the PCA of the environmental data, the time of day of the trial (am or pm), replicate testing (1 or 2), whether the control treatment was presented first (yes or no), and for the models based on the full dataset, treatment type (control or prey). To distinguish between predator responses to the prey stimulus within the apparatus and responses to the presentation of only a novel object (the control treatment), the analysis was conducted on the full dataset ($N_{Obs}$ = 228), and then repeated on subsets of the data based on treatment ($N_{Cont}$=114, $N_{Prey}$ = 114).

Model comparisons were based on the small-sample corrected Akaike Information Criterion (AICc). If a model had an AICc value smaller than the null model (by more than 2 units), it was considered strongly supported by the data[112]. When models fall within 0-2 units of the most likely model, it indicates a similar fit. However, if a larger model (i.e. with more parameters) is within this range of a smaller, most likely model, it suggests that the similarity in AICc values is primarily due to the addition of an extra parameter, rather than a genuine improvement in fit[113]. Using the 'DHARMa' v.0.4.6 package in R[105], each model was checked for the dispersion of the residuals. Models included the nested random effect of river and location to account for geographical differences; this consisted of 7 rivers and 57 locations. To improve the residual underdispersion and satisfy model assumptions for the presence/absence models in the prey treatment subset, these models included the random effect of river only.

## Data availability

Data supporting these findings is available under https://github.com/CosZan/Zanghi-et-al-2024-CommsBio.

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

## Acknowledgements
We thank the GW4 FRESH Centre for Doctoral Training in Freshwater Biosciences and Sustainability for their support in funding this project through an award to C.Z. (NE/R011524/1). Renoir Auguste and Bas Lloyd for their help and assistance while piloting the fieldwork. We would also like to thank the two anonymous referees for their critical and constructive feedback during the peer-review process.

## Author contributions
Contribution to experimental design: C.Z., I.P.W., C.C.I.; Data Collection: C.Z., data analysis: C.Z., with contributions from I.P.W., M.J.G., A.E.D., and C.C.I.; C.Z. wrote the first draft of the manuscript, and all authors contributed to revisions and approved the final version.

## Competing interests
The authors declare no competing interests.

## Ethical approval
All methods and procedures were performed in accordance with ASAB/ABS guidelines for the treatment of animals in behavioural research. The research was approved by the University Welfare and Ethical Review Body (UIN/ 21/003).
