## [Peer Review File · Communications Biology]

Reviewers' comments:

Reviewer #1 (Remarks to the Author):

Using field surveys and in situ presentation of stimulus prey shoals, the authors assess the impact of co-occurring environmental stressors (temperature and turbidity) on predator presence and activity. This manuscript represents a much needed addition to our understanding of how environmental variation impacts predator-prey interactions. As the authors mention, most previous laboratory and field research has focussed on prey species, leaving impacts on predator distribution under-examined. Overall, this is a well-written manuscript, and represents a well designed field survey. I have only a few (relatively minor) points to raise. Otherwise, this is an excellent piece of work and I look forward to seeing it in press!

Line 114-115: As I read the text, river width and depth were only recorded at one location per site. Would a single measurement be enough to capture the variability of these parameters within a site? Was width and depth recorded at the 'widest wetted reach' or 'max depth' or were they recorded at a haphazardly selected point? Some additional details are needed here.

Line 125: rather than 'predominant sediment type', would substrate diversity be a better predictor of the streambed. See Feyten et al. (2023) Ecology and Evolution.

Line 252: including the occurrence/behaviour of the two herbivore species (*H. robinii* and *A. maracase*) might be interesting. It is possible (likely?) that they may be using the presence of guppies as social foraging information. Such information would likely be impacted by the effects of temperature and turbidity, as shown for the predator community present.

Line 286-288: it is unclear what you mean by 'marginal effects', please clarify.

Line 290-298: I am curious if the authors found an effect of predator competition on the approach to the stimulus shoals? For example, Chris Elvidge found that the presence of pike cichlids reduced the approach rates of other common guppy predators.

Line 353-361: I am curious if the impacts of environmental stress would impact predator species differentially due to their primary sensory mode of foraging? Pike cichlids are more reliant on visual cues, while the blue acara and two-spot *Astyanax* use a combination of visual and chemosensory cues to localize prey (see work by Chris Elvidge). Additionally, aren't wolf fish primarily a nocturnal predator, possibly accounting for the low observed attack rate?

Line 373-375: I might also mention that future studies should also include aerial predators such as fishing bats and fishing spiders, as their predation efficiency would likely also be impacted by environmental stressors (especially turbidity).

Reviewer #2 (Remarks to the Author):

This is a fascinating paper presenting data from an experiment done in nature to understand how different predator species attack prey, and how various environmental factors play a role in the interaction. It is well designed, analyzed in a very competent way, and well written. I'm enthusiastic about this manuscript. I have some general comments that should be considered and have provided specific comments by line that I hope will enhance the manuscript.

General comments:

1. The Canonical Correspondence Analysis should be presented in more detail. The results from this analysis appear in the abstract and seem quite important, yet are summarized in single sentence in the results. I would like to see this treated more carefully, including where the 17% variance comes from. Ideally this would include a graph. I note that Supplementary Figure 5 was omitted from what was submitted, and this may address some of this concern.

2. A second general comment is that the authors should more clearly explain their approach to random effects. It seems like they use the words "field sites" and "locations" inconsistently, or at least unclearly. To me, the design seemed to have three random effects: river, field site nested in river, and location nested in field site nested in river. However, the authors seem to use only two random effects and I am somewhat confused. Whatever the analytical decision, it should be explained clearly.

3. My third and final general comment has to do with the experimental part of the study: having prey fish in the plastic container versus not. The treatment of this is key to understanding the results, yet the effect of this was unclearly presented. Some graphs comparing predator behavior between the prey versus no prey treatments would be much appreciated. I think that a clearer interpretation of these experimental treatments would also be beneficial. For example, in the treatment without prey fish, do the data tell you about what predators are and are not at each location? Combined with the environmental variables, does it tell about the abiotic niche of the different predators? Then a clear comparison between the prey/no prey treatments would really help illustrate how the predators respond to prey. I'm sure some of this is addressed in the manuscript, but it is not very clear.

Specific comments:

Abstract is well written and clear. My one suggestion would be to clarify what "predator community structure" is such that one can measure amount of variance explained in it. Is this species composition and abundance of predator species?

The introduction was also very well written and clear. It gave a nice overview of the environmental variables and predator-prey interactions, described the study system well, and provided testable hypotheses with predictions.

Methods:

L92-110: So far, what is unclear is whether there was a reasoning for having three sites per river and three locations per site, or whether this was just to have a reasonable sample size.

L118: I suggest stating what the instrument for turbidity is called. Is this specifically for measuring turbidity or a more generally used piece of equipment?

L152: How were top and side cameras synchronized? Normally, this would be done either electronically when they are activated, or using some clear event in the recordings.

L173: By "quarry", do the authors mean a rock quarry?

L190: should the random effects not be location nested in field site nested in river? There were three locations per field site and 19 field sites sampled from the seven rivers. Related to this, I am unclear if the authors changed the words used to describe each level of the design. Earlier, field sites were the 19 and locations were the three within each field site.

L194: The authors should justify why a zero inflation negative binomial analysis was used for time spent near stimulus. Did this model fit better than one excluding the zero inflation component? How was this determined?

L200: I would also suggest justifying and/or explaining the GLLVM, as this seems not a standard statistical technique that one can assume most are familiar with.

Results:

L238: Looking at Supp. Fig. 1, it is not obvious that these correlations would lead to any problems with multicollinearity, jeopardizing any analyses. In my experience, correlations can be useful for investigating which variables might be collinear if some variables are identified with low tolerances. I appreciate the authors' conservative approach, but think that they can get more out of their data here

if they would like to. I certainly leave this up to the authors. Furthermore, PCA does not assume that variables are not collinear (Discriminant Function Analysis, by contrast, does and is sensitive to it).

L246: I believe that the reference to PC1 in the middle of the line a typo (should be PC2).

L285-288: The statement about CCA is clear to me and I see that that is the result mentioned in the abstract. However, Supplementary figure 5 was not included in the supplementary material. This needs to be fixed, as it could be a critical figure, maybe even appropriate for the main manuscript.

L293-294: My interpretation of the Figure 2b is simply that as either PC increases, the number of species increases. As I type this, I realize that I am unclear if the response is number of species or presence/absence for any species (such that 0 predators =0 and 1+ predators =1). Either way, I do not really see evidence of a threshold, which I would imagine as a sudden switch from no to yes. I suppose I do see a gradual increase to a certain value of PC2/3 above which predators are always present, but is this a threshold?

L300-312: I find this a little confusing because of the replicate analyses with and without prey.

Wouldn't it be easier to do an analysis where prey versus control is a fixed effect?

Discussion:

- Overall, the discussion is interesting and clearly written. I think that a little more can be written comparing what other studies have found rather than limitations of the current study. I would also like to see a little more on the interpretation of the prey-free data. For example, does this tell you more about what predators exist at each site? Does that tell you about abiotic niche use of predators?

L324-337: Although it is fine to consider limitations of the current study, I suggest leading the discussion with what the results' implications are, maybe starting with the most impactful parts. Leave the limitations until near the end.

L328: Please define NTU. Perhaps this is better on line 118 in the methods.

L338: Considering an earlier comment, I actually can't point to a clear result that shows this lack of treatment effect. I know it is in the results, but since that is a considerable focus of this study, it would be great to see a figure that shows this result.

Table 2: I like the key for helping to interpret/remember PC1, 2, and 3. What I would add to this is some indication of how each variable is related to each PC, for example, positively or negatively.

Figure 2: For both graphs I suggest including the data points on the plot.

Figure 3: I note that the graphs of community composition use count on the y-axis. Does changing these to proportion, such that the stacked bars all go to 1 (100%) change your interpretation? The current graphs mainly tell you that predators are much more abundant in St. Joseph R. or on leaf litter or pebbles than other places. Bars scaled to uniform height would give more of a view of relative abundances.

Figure 4: Currently this one is also labeled figure 2. I like this figure, although suggest using somewhat bolder colors for ease of viewing (color is not strictly needed for this figure).

Supplementary Table 4: This table is rather redundant with Figure 3. This is okay, but I'd appreciate maybe some more output from the GLLVM analyses, like for example, how much variation was explained by river or field site.

Supplementary Table 5: "dAICc" should be changed to the Greek letter delta because that is convention.

Supplementary Figures 2 & 3: It would be good to include these statistics in the methods (that these variables were compared across rivers with Kruskal-Wallis tests, followed by Dunn's tests, with the Benjamini-Hochberg correction for multiple comparisons).

Supplementary Figure 4: Earlier graphs arranged the rivers from West to East, but this one does it

from largest to smallest guppy. This might lead to confusion and I suggest arranging this figure as the others were.

Supplementary Figure 5: Missing.

10th April 2024

Dear Reviewers,

Thank you for your interest in our manuscript (COMMSBIO-24-0712) entitled “Multiple environmental stressors affect predation pressure in a tropical freshwater system”. We greatly appreciate your efforts, positivity, and suggestions to improve this work.

Please find below our responses to the comments that have arisen from the reviewing process. We have attempted to respond thoroughly and clearly to all comments. As part of the resubmission, we have uploaded a ‘tracked changes’ version of the original manuscript and supplementary document. We have also uploaded a ‘clean’ version of both files where all changes have been accepted and fully integrated in the text. The line numbers mentioned in the responses below refer to the clean versions.

Once again, thank you for your valued help at this stage.

Sincerely,

The authors

Reviewers' comments:

Reviewer #1 (Remarks to the Author):

Using field surveys and in situ presentation of stimulus prey shoals, the authors assess the impact of co-occurring environmental stressors (temperature and turbidity) on predator presence and activity. This manuscript represents a much needed addition to our understanding of how environmental variation impacts predator-prey interactions. As the authors mention, most previous laboratory and field research has focussed on prey species, leaving impacts on predator distribution under-examined. Overall, this is a well-written manuscript, and represents a well designed field survey. I have only a few (relatively minor) points to raise. Otherwise, this is an excellent piece of work and I look forward to seeing it in press!

Line 114-115: As I read the text, river width and depth were only recorded at one location per site. Would a single measurement be enough to capture the variability of these parameters within a site? Was width and depth recorded at the ‘widest wetted reach’ or ‘max depth’ or were they recorded at a haphazardly selected point? Some additional details are needed here.

Response: We have specified how the points were selected for these measurements, details on L117-119. Importantly, we have clarified that the locations of the width and depth measurements are where the behavioural observations were carried out.

Line 125: rather than ‘predominant sediment type’, would substrate diversity be a better predictor of the streambed. See Feyten et al. (2023) Ecology and Evolution.

Response: This is a good point, but considering the small area sampled for sediment type (to correspond to where the behavioural observation took place – not to be treated as a predictor of habitat diversity) we believe our method is appropriate.

Line 252: including the occurrence/behaviour of the two herbivore species (*H. robinii* and *A. maracase*) might be interesting. It is possible (likely?) that they may be using the presence of guppies

as social foraging information. Such information would likely be impacted by the effects of temperature and turbidity, as shown for the predator community present.

Response: This is a very interesting point; for this study we focused only on predator-prey interactions, however we agree it would be interesting to include other inter-specific social interactions or also social interactions between free-roaming and stimulus guppies. We have included considerations of this in the discussion section (L382-385)

Line 286-288: it is unclear what you mean by 'marginal effects', please clarify.

Response: We have removed the term "marginal" due to its lack of clarity (L303).

Line 290-298: I am curious if the authors found an effect of predator competition on the approach to the stimulus shoals? For example, Chris Elvidge found that the presence of pike cichlids reduced the approach rates of other common guppy predators.

Response: We have not looked at predators' interactions and competition, as this is out of the scope of the current study which was designed to determine the effects of environmental variables, not the effect of predator species on one another. To look at interactions between predators, the observations would need to be paired with larger-scale data on the abundances of each predator species e.g. within a pool based on electrofishing. Then the presence of each species near the prey stimulus would need to be analysed controlling for the local abundance of each species. Anecdotally, in our videos we observed direct aggression from pike cichlids to other pike cichlids and to blue acaras, but no aggression from blue acaras to pike cichlids. We agree this is an interesting point and it is something we would be interested in exploring in the future. Similarly to the previous comment about other social interactions, we have expanded the discussion to cover these points (L382-385).

Line 353-361: I am curious if the impacts of environmental stress would impact predator species differentially due to their primary sensory mode of foraging? Pike cichlids are more reliant on visual cues, while the blue acara and two-spot *Astyanax* use a combination of visual and chemosensory cues to localize prey (see work by Chris Elvidge). Additionally, aren't wolf fish primarily a nocturnal predator, possibly accounting for the low observed attack rate?

Response: We focused our study on visual predators (e.g. L269-270). However, the cylinder containing the guppies was not sealed, therefore we assume there was some odour cue as well as visual. We have added more detail on this in the discussion (L373, 375). We agree that the wolf fish is considered predominantly a crepuscular/nocturnal predator, we did however encounter it regularly during our observations in the daytime, and when observed it always performed at least one attack; we suspect that the relatively low attack rate is due to its stalking behaviour (long time near stimulus/low number of attacks) rather than low incidence of individuals (Table 1).

Line 373-375: I might also mention that future studies should also include aerial predators such as fishing bats and fishing spiders, as their predation efficiency would likely also be impacted by environmental stressors (especially turbidity).

Response: this is another good point, we have included this suggestion on L375.

Reviewer #2 (Remarks to the Author):

This is a fascinating paper presenting data from an experiment done in nature to understand how different predator species attack prey, and how various environmental factors play a role in the interaction. It is well designed, analyzed in a very competent way, and well written. I'm enthusiastic

about this manuscript. I have some general comments that should be considered and have provided specific comments by line that I hope will enhance the manuscript.

General comments:

1. The Canonical Correspondence Analysis should be presented in more detail. The results from this analysis appear in the abstract and seem quite important, yet are summarized in single sentence in the results. I would like to see this treated more carefully, including where the 17% variance comes from. Ideally this would include a graph. I note that Supplementary Figure 5 was omitted from what was submitted, and this may address some of this concern.

Response: We have added more detail on the results of the CCA within the text (L300-302), referring earlier on to Supplementary Figure 5 (now included in the document).

2. A second general comment is that the authors should more clearly explain their approach to random effects. It seems like they use the words “field sites” and “locations” inconsistently, or at least unclearly. To me, the design seemed to have three random effects: river, field site nested in river, and location nested in field site nested in river. However, the authors seem to use only two random effects and I am somewhat confused. Whatever the analytical decision, it should be explained clearly.

Response: This is a good point; we have explained in the specific comment below (Q: L190) how and why the random effects were selected. We have also added more detail in the manuscript to avoid further confusion, see L198-200.

3. My third and final general comment has to do with the experimental part of the study: having prey fish in the plastic container versus not. The treatment of this is key to understanding the results, yet the effect of this was unclearly presented. Some graphs comparing predator behavior between the prey versus no prey treatments would be much appreciated. I think that a clearer interpretation of these experimental treatments would also be beneficial. For example, in the treatment without prey fish, do the data tell you about what predators are and are not at each location? Combined with the environmental variables, does it tell about the abiotic niche of the different predators? Then a clear comparison between the prey/no prey treatments would really help illustrate how the predators respond to prey. I’m sure some of this is addressed in the manuscript, but it is not very clear.

Response: We agree that more attention to the effects of treatment type on the behaviours observed was necessary, and we have now included more detail in the text (L314-316), added a column on Figure 4 for treatment effect for each behavioural metric, and included a new figure in the supplementary file with presence/absence by treatment (details of these are found in our answers to the specific comments below).

Regarding more interpretation of the prey-free data, after careful consideration, we do not believe it would be appropriate for this study to speculate on those results (e.g. predators’ abiotic niches). The prey-free treatment was designed to assess the impact of a novel object and to be compared to predators’ behaviours in relation to the prey stimulus as a way to validate the experimental design, but we are cautious in extrapolating more from such data. We would need further sampling, for example electro-fishing, traditional seining surveys or eDNA assessments to ground-truth the data obtained by video recording and generalise on the behaviours and space-use of these predators when we are not presenting prey. Nonetheless, we understand this to be a limitation of the current experimental design and it has been mentioned in the discussion (L384) also in response to comments from reviewer 1.

Specific comments:

Abstract is well written and clear. My one suggestion would be to clarify what “predator community structure” is such that one can measure amount of variance explained in it. Is this species composition and abundance of predator species?

Response: This has been amended (L17).

The introduction was also very well written and clear. It gave a nice overview of the environmental variables and predator-prey interactions, described the study system well, and provided testable hypotheses with predictions.

Methods:

L92-110: So far, what is unclear is whether there was a reasoning for having three sites per river and three locations per site, or whether this was just to have a reasonable sample size.

Response: This has now been specified on L96.

L118: I suggest stating what the instrument for turbidity is called. Is this specifically for measuring turbidity or a more generally used piece of equipment?

Response: This has now been added, alongside the specification of NTU (L122-123).

L152: How were top and side cameras synchronized? Normally, this would be done either electronically when they are activated, or using some clear event in the recordings.

Response: We used a hand signal to manually synchronise the videos, we have slightly re-worded the sentence to improve clarity (L157).

L173: By “quarry”, do the authors mean a rock quarry?

Response: Correct, this has been clarified (L180).

L190: should the random effects not be location nested in field site nested in river? There were three locations per field site and 19 field sites sampled from the seven rivers. Related to this, I am unclear if the authors changed the words used to describe each level of the design. Earlier, field sites were the 19 and locations were the three within each field site.

Response: We apologise for the confusion caused by our wording; we have now specified we did not include Site (i.e. field site) in the models. Due to the hierarchical nature of the sampling units, by including the random effect of location within river, we are already accounting for the variability associated with different sampling points within each river. Including the site as an additional random effect might introduce redundancy in the models because variation in the site may be captured by the combination of river and location already. Additionally, a nested random effect with 3 levels would increase model complexity and potentially causing convergence issues. We have clarified this as we understand it is important to fully explain the reasoning behind models' terms selection (L198-200).

L194: The authors should justify why a zero inflation negative binomial analysis was used for time spent near stimulus. Did this model fit better than one excluding the zero inflation component? How was this determined?

Response: We now specify that zero inflation was added to this model to solve convergence issues (L202-203).

L200: I would also suggest justifying and/or explaining the GLLVM, as this seems not a standard statistical technique that one can assume most are familiar with.

Response: We agree that this analysis needs more detail. We have now added reasoning and the model selection process on L209-211 and 214-216.

Results:

L238: Looking at Supp. Fig. 1, it is not obvious that these correlations would lead to any problems with multicollinearity, jeopardizing any analyses. In my experience, correlations can be useful for investigating which variables might be collinear if some variables are identified with low tolerances. I appreciate the authors' conservative approach, but think that they can get more out of their data here if they would like to. I certainly leave this up to the authors. Furthermore, PCA does not assume that variables are not collinear (Discriminant Function Analysis, by contrast, does and is sensitive to it).

Response: We agree with the reviewer's comments on this point. The only variable that we did not include in the behavioural analysis due to multicollinearity issues with the other predictors was sediment type (Supplementary Figure 3). We decided to take a PCA approach to simplify the behavioural modelling analysis as we state on L184 "To reduce the dimensionality of the environmental data". We did not use PCA assuming that the variables are not collinear, rather we used PCA to reduce the number of predictor variables in the behavioural analysis, and avoid possible multicollinearity within these models.

L246: I believe that the reference to PC1 in the middle of the line a typo (should be PC2).

Response: We can confirm that PC1 is correct (L259), this is shown in Supplementary Table 2, and Figure 3 and 4 where light intensity, river width and canopy cover are associated with PC1.

L285-288: The statement about CCA is clear to me and I see that that is the result mentioned in the abstract. However, Supplementary Figure 5 was not included in the supplementary material. This needs to be fixed, as it could be a critical figure, maybe even appropriate for the main manuscript.

Response: We apologise for the oversight regarding the omission of Supplementary Figure 5 and appreciate it being brought to our attention. Given the manuscript's current content, which already includes four multi-panel figures alongside tables, we find Supplementary Figure 5 more suitable in the supplementary information. Nevertheless, we remain open to its inclusion in the main manuscript if deemed appropriate by the editor.

L293-294: My interpretation of the Figure 2b is simply that as either PC increases, the number of species increases. As I type this, I realize that I am unclear if the response is number of species or presence/absence for any species (such that 0 predators =0 and 1+ predators =1). Either way, I do not really see evidence of a threshold, which I would imagine as a sudden switch from no to yes. I suppose I do see a gradual increase to a certain value of PC2/3 above which predators are always present, but is this a threshold?

Response: The figure represents the model outcome for the predicted relationship between PC2 and PC3 for presence/absence (0/1) for any species, not number of species. We agree that the change is more gradual rather than a sudden switch, therefore we have reworded the term "threshold" from our description (L309-310). We have also clarified in the text that in the multivariate analysis

presence/absence data is per species (i.e. multiple values (0/1) per sampling event) on L208, while for the behavioural analysis presence/absence refers to any predator species (i.e. one value: 0 = no predators, 1 = any predator per sampling event) on L227.

L300-312: I find this a little confusing because of the replicate analyses with and without prey. Wouldn't it be easier to do an analysis where prey versus control is a fixed effect?

Response: As shown in Supplementary Table 5, the analysis performed on the full dataset included Treatment type as a fixed effect to assess the influence of treatment on the behavioural responses. However, we decided to repeat the analysis on the separate treatment types to allow for increased sensitivity to more subtle effects and targeted interpretation of behavioural responses to the presence of prey specifically. To improve clarity, we have specified when we refer to the prey treatment dataset and when we refer to the full dataset L 320, 324, and 325-327.

Discussion:

- Overall, the discussion is interesting and clearly written. I think that a little more can be written comparing what other studies have found rather than limitations of the current study. I would also like to see a little more on the interpretation of the prey-free data. For example, does this tell you more about what predators exist at each site? Does that tell you about abiotic niche use of predators?

Response: We agree with the point made here and in the following comment and we have re-structured the discussion section to highlight the actual results first (the paragraph on L386 was moved from L343). Regarding the interpretation of the prey-free data, we have given justification on this in the answer to general question #3.

L324-337: Although it is fine to consider limitations of the current study, I suggest leading the discussion with what the results' implications are, maybe starting with the most impactful parts. Leave the limitations until near the end.

Response: We agree with this comment and have re-shuffled the section accordingly, see L386.

L328: Please define NTU. Perhaps this is better on line 118 in the methods.

Response: The definition of NTU has been added in the method section as suggested.

L338: Considering an earlier comment, I actually can't point to a clear result that shows this lack of treatment effect. I know it is in the results, but since that is a considerable focus of this study, it would be great to see a figure that shows this result.

Response: This is a good point, we have added the specific result referring to the supplementary table on L314-316, we have added boxplots of the treatment effect for each behavioural metric (Figure 4 column 4), and we have now included a new Supplementary Figure (6) with presence/absence for the control and prey treatments.

Table 2: I like the key for helping to interpret/remember PC1, 2, and 3. What I would add to this is some indication of how each variable is related to each PC, for example, positively or negatively.

Response: +/- signs have now been added alongside each variable associated with each principal component, according to Supplementary Table 2.

Figure 2: For both graphs I suggest including the data points on the plot.

Response: After consideration, Figure 2a would be crowded and difficult to read with the addition of the data points, so we have decided to retain it as it is. However we agree that the addition of datapoints in Figure 2b would improve it, so the amended plot has now been included.

Figure 3: I note that the graphs of community composition use count on the y-axis. Does changing these to proportion, such that the stacked bars all go to 1 (100%) change your interpretation? The current graphs mainly tell you that predators are much more abundant in St. Joseph R. or on leaf litter or pebbles than other places. Bars scaled to uniform height would give more of a view of relative abundances.

Response: We have amended the figure so percentages instead of count data is now displayed. The text has been changed accordingly in the figure caption and on L277-279.

Figure 4: Currently this one is also labeled figure 2. I like this figure, although suggest using somewhat bolder colors for ease of viewing (color is not strictly needed for this figure).

Response: Thank you for pointing this out, figure numbers have been checked throughout. We have modified the palette of all figures to be consistent throughout and we have assured the colours chosen are fully accessible for all types of colour blindness through the tool ColorOracle (<https://colororacle.org/>).

Supplementary Table 4: This table is rather redundant with Figure 3. This is okay, but I'd appreciate maybe some more output from the GLLVM analyses, like for example, how much variation was explained by river or field site.

Response: We have removed the detail in the methods about "at the field sites" on L208, as we understand this to be misleading. The GLLVM doesn't assess variation/influence of the geographical location on the presence/absence of the species, but it considers the environmental conditions found at the sites.

Supplementary Table 5: "dAICc" should be changed to the Greek letter delta because that is convention.

Response: This has been amended.

Supplementary Figures 2 & 3: It would be good to include these statistics in the methods (that these variables were compared across rivers with Kruskal-Wallis tests, followed by Dunn's tests, with the Benjamini-Hochberg correction for multiple comparisons).

Response: This has been added on L177-179.

Supplementary Figure 4: Earlier graphs arranged the rivers from West to East, but this one does it from largest to smallest guppy. This might lead to confusion and I suggest arranging this figure as the others were.

Response: We agree with this comment, the figure has been modified accordingly.

Supplementary Figure 5: Missing.

Response: We apologise for this oversight and thank the reviewer for pointing out this mistake. This has now been included.

REVIEWERS' COMMENTS:

Reviewer #1 (Remarks to the Author):

Having carefully reviewed the authors' responses to my initial comments (and those of the other reviewer), I'm happy to recommend acceptance of the manuscript in its revised form. The authors have satisfactorily dealt with my concerns. I look forward to seeing the paper in print!

Reviewer #2 (Remarks to the Author):

I have re-read the manuscript and the responses to reviewer comments, and think that the authors did a great job addressing the feedback. I have only one final comment for consideration:

- In the new figure 2b, it is hard to see the open white circles of the data. Would it be possible to change them to something bolder? Yellow and white are not a great combination due to low contrast between them. Black could be an option, but would be low contrast with the purple. However, this would be preferable because it would make only one or two dots hard to see rather than the majority.